# A single main-chain hydrogen bond required to keep GABA_A receptors closed

Cecilia M. Borghese [1], Jason D. Galpin [2], Samuel Eriksson Lidbrink [3], Yuxuan Zhuang [3], Netrang G. Desai[1], Rebecca J. Howard [3,4], Erik Lindahl [3,4], Christopher A. Ahern [2] & Marcel P. Goldschen-Ohm [1] ✉

GABA_A receptors (GABA_ARs) are the primary inhibitory neurotransmitter receptors throughout the central nervous system. Genetic mutations causing their dysfunction are related to a broad spectrum of human disorders such as epilepsy, neurodevelopment and intellectual disability, autism spectrum disorder, schizophrenia, and depression. GABA_ARs are also important drug targets for anxiolytics, anticonvulsants, antidepressants, and anesthetics. Despite significant progress in understanding their three-dimensional structure, a critical gap remains in determining the molecular basis for channel gating. We recently identified mutations in the M2-M3 linkers that suggest linker flexibility has asymmetric subunit-specific correlations with channel opening. Here we use non-canonical amino acids (ncAAs) to investigate the role of main-chain H-hydrogen bonds (H-bonds) that may stabilize the M2-M3 linkers. We show that a single main-chain H-bond within the β2 subunit M2-M3 linker inhibits pore opening and is required to keep the unliganded channel closed. Furthermore, breaking this H-bond accounts for approximately one third of the energy used to open the channel during activation by GABA. In contrast, the analogous H-bond in the α1 subunit has no effect on gating. Our molecular simulations support the idea that channel opening involves the state-dependent breakage/ disruption of a specific main-chain H-bond within the β2 subunit M2-M3 linker.

Main-chain H-bonds are essential for proteins to fold into helix and sheet secondary structures. Despite their obvious role in structure, their mechanistic contribution to the function of mature proteins is typically unknown, especially in less ordered loops where their importance is often not structurally obvious. The relative lack of functional measures that probe main-chain H-bonds is largely due to the fact that conventional site-directed mutagenesis only allows swapping side-chains but does not alter the composition of the main-chain. Nonetheless, there exist well-established methods for introducing ncAAs with altered main-chain chemistry[1,2]. In particular, α-hydroxy acids enable amide-to-ester substitutions in the main-chain that ablate H-bonds with the amide nitrogen while leaving the side-chain unchanged[1,3]. Effectively, this enables targeted elimination of specific main-chain H-bonds to test their contribution to protein function.

Although largely under-utilized for ion channels, incorporation of α-hydroxy acids has revealed important roles for main-chain H-bonds in ion selectivity of acid-sensing (ASIC) and chloride (CLC) channels[4,5], voltage-sensitivity of Shaker potassium channels[6], gating dynamics for inwardly rectifying potassium (Kir) channels[7,8], both ligand binding and channel gating in nACh receptors[9–15], and stability of the open state in the prokaryotic homolog GLIC[16]. Here, we determine the mechanistic contribution of a main-chain H-bond within an important gating loop of a typical synaptic GABA_AR.

[1]Department of Neuroscience, University of Texas at Austin, Austin, TX, USA. [2]Department of Molecular Physiology and Biophysics, University of Iowa, Iowa City, IA, USA. [3]Department of Biochemistry and Biophysics, Stockholm University, Stockholm, Sweden. [4]Department of Applied Physics, KTH Royal Institute of Technology, Stockholm, Sweden. ✉e-mail: marcel.goldschen-ohm@austin.utexas.edu

GABA$_A$Rs are members of the cys-loop superfamily of pentameric ligand-gated ion channels (pLGICs) including glycine (Gly), nicotinic acetylcholine (nACh), and serotonin (5-HT3) receptors[17]. They are comprised of subtype-specific combinations of homologous but nonidentical subunits (α1–6, β1–3, γ1–3, δ, ε, π, θ, ρ1–3) that together form a central chloride-conducting pore through the plasma membrane[18–20]. Cryo-EM structures of common synaptic subtypes α1β2/3γ2 GABA$_A$Rs have been enormously useful for mechanistic inference and prediction (Fig. 1a)[20–29]. Agonists (e.g., the neurotransmitter GABA) bind to two sites in the extracellular domain (ECD) at β/α inter-subunit interfaces, where they promote opening of the ion pore in the transmembrane domain (TMD). Other distinct sites mediate allosteric modulation by anxiolytics (e.g., diazepam), anticonvulsants, antidepressants, and anesthetics. Loops at the ECD-TMD coupling interface including the M2-M3 linkers are crucial for transducing the chemical energy from agonist binding to pore gating (Fig. 1b)[30–33]. However, the detailed interactions mediating this coupling remain only poorly understood. Comparison of structural snapshots in inactive (e.g., antagonist-bound) and activated/desensitized (e.g., GABA-bound) conformations suggest a prominent motion during agonist-activation is a radial expansion of the M2-M3 linkers, which directly follow the pore-lining M2 helices in the polypeptide sequence[23,34,35].

Our previous observations that alanine substitutions predicted to increase M2-M3 linker flexibility have highly asymmetric subunit-specific effects on gating and diazepam modulation[36,37] led us to explore other strategies to increase M2-M3 linker flexibility. Structures indicate a potential main-chain H-bond within the M2-M3 linker that could stabilize a crimp or shallow hairpin in the linker backbone predicted to limit linker flexibility (Fig. 1b).

Here, we show that this H-bond in the β2 subunit is a crucial component of the channel gating mechanism, whereas it has no functional role in the α1 subunit despite high structural homology.

## Results

### Main-chain H-bond elimination with ncAAs

To eliminate the putative main-chain H-bonds α1(Val279:N-Pro277:O) or β2(Ile275:N-Pro273:O) within the M2-M3 linkers (Fig. 1b), either α1(Val279) or β2(Ile275) was substituted with the amber stop codon (TAG), denoted α1(Val279*) or β2(Ile275*). We then used in vivo nonsense suppression to introduce the cognate α-hydroxy acid (Vah or Iah) at the cRNA UAG site as previously described[6]. The amide-to-ester substitutions α1(Val279Vah) or β2(Ile275Iah) effectively eliminate any main-chain H-bonding between the amide nitrogen of α1(Val279) or β2(Ile275) and the carboxylate oxygen of α1(Pro277) or β2(Pro273), respectively, without altering the side-chains of these residues (Figs. 1b, 2a). Briefly, *Xenopus laevis* oocytes were coinjected with cRNA for α1, β2, and γ2 subunits (or cRNA for specific UAG mutants) and an orthogonal pyrrolysine tRNA (a natural UAG suppressing tRNA) that has been chemoenzymatically ligated with either i) the cognate α-hydroxy acid (the test case), ii) the wild-type (WT) amino acid (should recapitulate WT behavior), or iii) nothing (blank full-length tRNA; should not express GABA$_A$R due to early termination at the introduced stop codon) (Fig. 2b). For each batch of oocytes, current responses to pulses of the pore blocker picrotoxin (PTX) and a series of GABA concentrations were recorded from oocytes in all three conditions (Fig. 2c). Reliable nonsense suppression incorporation of residues with little to no read-through is evident from the lack of GABA$_A$R currents from oocytes in condition iii as compared to conditions i and ii (Fig. 2d) and the recapitulation of WT behavior in condition ii (Fig. 3).

### H-bond in β2 inhibits pore opening

We initially expected rather small functional effects from ablating individual H-bonds. Thus, we first evaluated these perturbations in the α1(Leu9′Thr)β2γ2 gain-of-function (GoF) background (Leu9′ corresponds to Leu263 or Leu264 in the mature rat or human α1 subunit, respectively, see Supplementary Fig. 1). The substitution α1(Leu9′Thr) in the main pore gate stabilizes the open state such that channels open spontaneously in the absence of agonist with appreciable probability[38]. This behavior allows relatively small changes in the closed-open equilibrium to be readily observed as changes in ionic current[36,37]. Due to limited desensitization, the peak response of GoF receptors to saturating GABA is predicted to be close to maximal (i.e., an open probability of 1). Application of the pore blocker PTX blocks any spontaneously open channels and reveals the zero-current baseline (i.e., an open probability of 0). Thus, we estimated channel open probability by normalizing currents between the extreme levels elicited separately with PTX and saturating GABA ($I_{total}$) (Figs. 2c, 3a). Basal unliganded open probability ($P_o$) is thus given by the ratio of PTX-sensitive to total current amplitude ($I_{PTX}/I_{total}$), from which the closed-open free energy difference is computed as

$$\Delta G = -RT\ln\left(\frac{P_o}{1-P_o}\right) \quad (1)$$

where $R$ is the gas constant and $T$ is temperature.

Nonsense suppression reintroduction of WT amino acids at TAG sites, α1(Val279Val) or β2(Ile275Ile), recapitulated GoF behavior, including both basal $P_o$ (3a–c left, middle) and GABA sensitivity (Fig. 3d left, middle). Although there was a slight shallowing of the GABA concentration-response curves (CRCs) for nonsense suppression incorporation of residues at β2(Ile275*) as compared to GoF, this effect was minor in comparison to the effects we will focus on. Similarly, ablation of the predicted H-bond α1(Val279:N-Pro277:O) with the

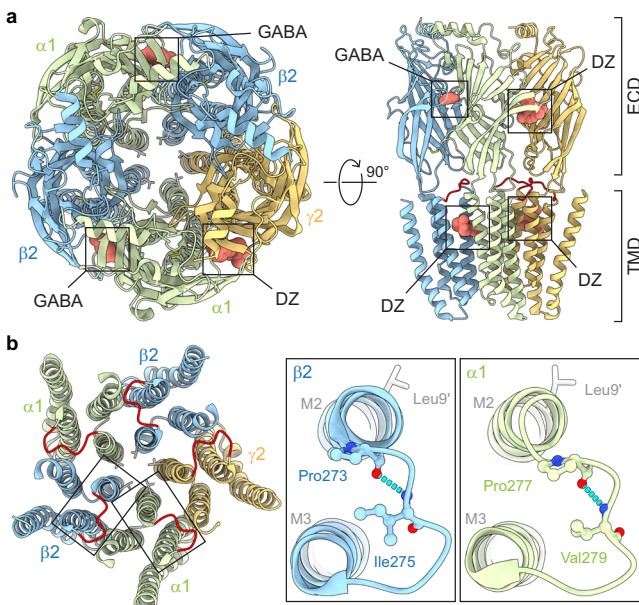

**Fig. 1 | Predicted main-chain H-bonds within M2-M3 linkers. a** Cryo-EM map (PDB 6X3X) of human α1β2γ2 GABA$_A$R with bound GABA and the positive modulator diazepam (DZ) viewed from the top (left) and side (right) (ECD, extracellular domain). M2-M3 linkers colored red on right. **b** Left: Top view of transmembrane domain (TMD) only for antagonist-bound structure (PDB 6X3S). M2-M3 linkers colored red and 9′ pore gate leucines shown as sticks. Right: M2-M3 linkers for boxed regions on left. Predicted main-chain H-bonds α1(Val279:N-Pro277:O) and β2(Ile275:N-Pro273:O) indicated with cyan dashed segments and involved residues shown as ball and stick. Rat residue numbering shown, which is offset by one from human numbering in α1 (Supplementary Fig. 1). A similar H-bond is predicted in the γ2 subunit (Val290:N-Pro288:O). Structure visualizations with ChimeraX[46].

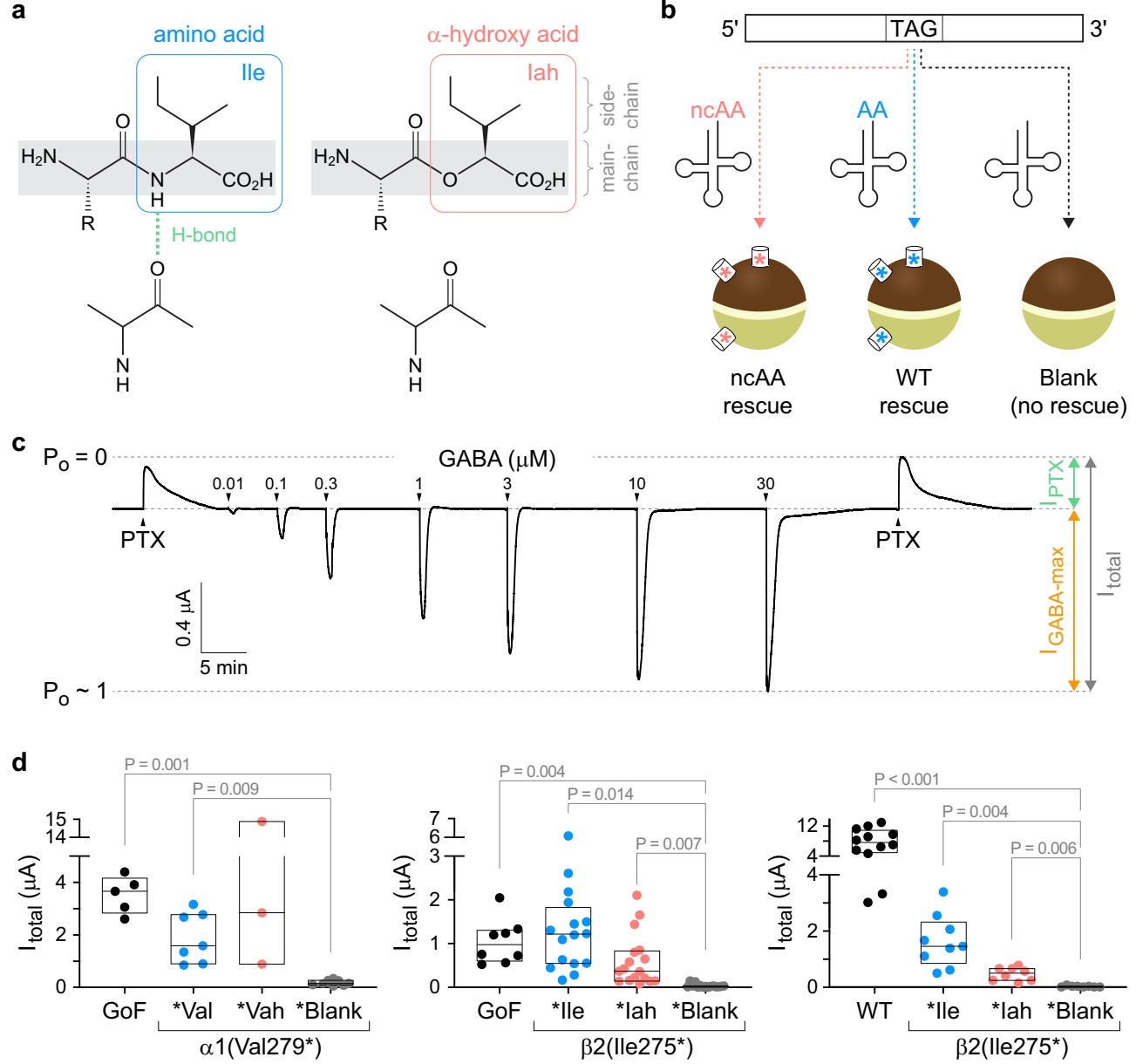

**Fig. 2 | Nonsense suppression incorporation of α-hydroxy acids to eliminate specific main-chain H-bonds. a** Depiction of a main-chain H-bond with the amide NH of an isoleucine (Ile) amino acid residue and its ablation upon substitution with its cognate α-hydroxy acid (Iah). **b** Oocytes are coinjected with cRNA for GABA$_A$R subunits with a TAG stop codon at the site of interest along with tRNA ligated to either the wild-type amino acid (AA), its cognate α-hydroxy acid (ncAA), or nothing (blank). The result is expression of wild-type channels, channels incorporating the α-hydroxy acid at the TAG site, or no channels due to truncation at the TAG site, respectively. **c** Current trace for α1(Leu9'Thr)β2γ2 gain-of-function (GoF) receptors. Arrows indicate approximate onset of pulses of either 1 mM picrotoxin (PTX) or increasing concentrations of GABA. Open probability ($P_o$) estimated by normalizing from the zero-current baseline in PTX to the maximal current elicited with GABA. **d** Total current per oocyte as illustrated in panel (**c**) (left to right: $n = 5$, 7, 3, 8; 8, 16, 17, 15; 12, 9, 8, 8) suggests reliable nonsense suppression incorporation of AA and ncAA with little to no read-through (i.e., relative lack of current for blank) in both GoF and wild type (WT) backgrounds. For example, the notation Val279* denotes a TAG stop codon at position 279, and *Val or *Vah imply nonsense suppression incorporation of Val or Vah at the TAG site resulting in Val279Val or Val279Vah, respectively. *Blank indicates an unchanged TAG stop codon. Box plots show median and interquartile intervals. $P$-values < 0.05 for Brown-Forsythe ANOVA with posthoc Dunnett's T3 test shown.

substitution α1(Val279Vah) had no effect on GoF behavior, suggesting that either this H-bond does not exist, or it is irrelevant to channel function. Given the location of the α1 subunit M2-M3 linker below the classical benzodiazepine site in the ECD and our previous observation that α1(Val279Ala) enhances diazepam modulation[36], we asked whether the α1(Val279:N-Pro277:O) H-bond was involved in allosteric modulation by benzodiazepines such as diazepam. Consistent with its lack of effect on GABA-elicited currents, α1(Val279Vah) also had no effect on allosteric modulation by diazepam (Supplementary Fig. 3).

In contrast, ablation of the predicted H-bond β2(Ile275:N-Pro273:O) with the substitution β2(Ile275Iah) increased the estimated basal $P_o$ by ~0.6 as compared to GoF (Fig. 3a, b middle), suggesting that this H-bond naturally inhibits channel opening by ~1.8 kcal/mol (Fig. 3c middle). Despite the increase in $P_o$, we did not observe an associated increase in GABA sensitivity (i.e., left-shift of the CRC) (Fig. 3d middle). The reason for this is unclear, but could in part reflect overlapping mechanisms for the sensitizing effects of the GoF mutation α1(Leu9'Thr) and the H-bond ablation β2(Ile275Iah), which individually both left-shift the GABA CRC

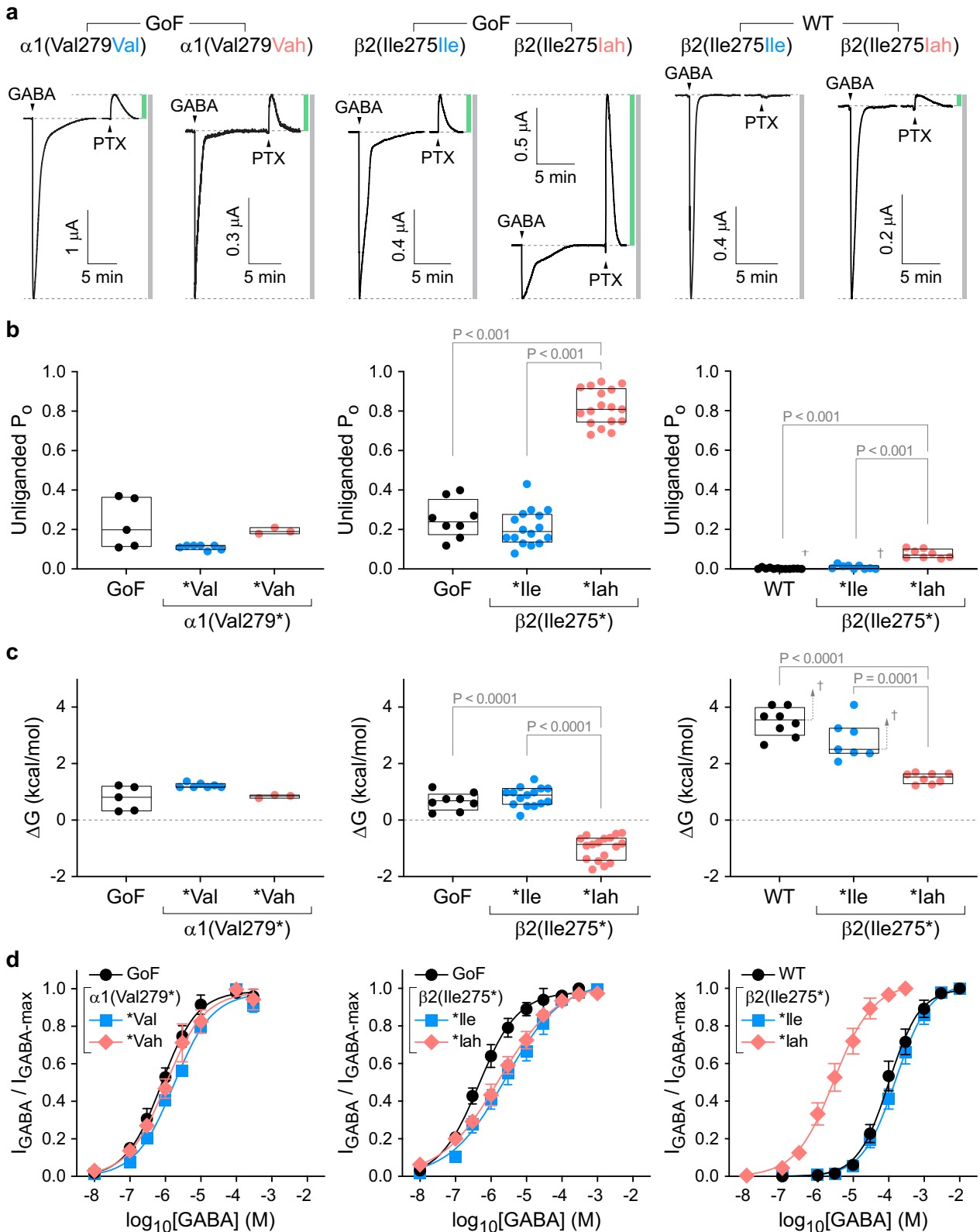

(Fig. 3d). In summary, these results suggest that the H-bond β2(Ile275:N-Pro273:O) inhibits channel opening, whereas the analogous H-bond α1(Val279:N-Pro277:O) does not contribute to channel gating.

## H-bond in β2 required to stay closed

Based on the large effect of β2(Ile275Iah) in the GoF background, we examined substitutions at β2(Ile275*) in the α1β2γ2 WT background.

First, β2(Ile275Ile) recapitulated WT behavior including little to no PTX-sensitive basal current (3a–c right) and normal GABA sensitivity (Fig. 3d right). Consistent with our observation in the GoF background, ablation of the predicted H-bond β2(Ile275:N-Pro273:O) in a WT background with the substitution β2(Ile275Iah) increased the basal $P_o$ to the extent that channels were spontaneously open ~7% of the time (Fig. 3a, b right). Although we cannot rule out an effect on

**Fig. 3 | The β2 subunit main-chain Ile275:N-Pro273:O H-bond inhibits pore opening and is required to keep the unliganded channel closed. a** Current responses to 10–20 s pulses (arrows) of saturating GABA or 1 mM picrotoxin (PTX) for α1β2γ2 (WT) or α1(Leu9′Thr)β2γ2 (GoF) receptors after nonsense suppression incorporation of either the wild-type amino acid (Val, Ile) or its cognate α-hydroxy acid (Vah, Iah) at α1(Val279*) or β2(Ile275*). Vertical bars indicate the relative magnitude of PTX-sensitive (green) to total (gray) currents. **b** Unliganded open probability ($P_o$) per oocyte estimated as the fraction of PTX-sensitive to total current (left-to-right: $n$ = 5, 7, 3; 8, 16, 17; 12, 9, 8). For example, the notation Ile275* denotes a TAG stop codon at position 275, and *Ile or *Iah imply nonsense suppression incorporation of Ile or Iah at the TAG site resulting in Ile275Ile or Ile275Iah, respectively. *Blank indicates an unchanged TAG stop codon. Box plots show median and interquartile intervals. $P$-values < 0.05 for Brown-Forsythe ANOVA with posthoc Dunnett's T3 test shown. Values for GoF in the left or middle panels reflect oocytes from the same batches as the other recordings in each respective panel. **c** Free energy difference between closed and open states in the absence of ligand for each oocyte computed from the estimated open probabilities in panel b. Box plots and P-values as in panel (**b**). **d** Normalized concentration-response relations for GABA-elicited currents. Data are mean ± SEM across oocytes. Curves are the Hill equation fit to the means (Eq. 2). See Supplementary Tables 1, 2 for summary statistics and fit parameters [n per condition: left panel: GoF, 5; α1(Val279Vah), 7; α1(Val279Val), 6; middle panel: GoF, 5; β2(Ile275Ile), 5; β2(Ile275Iah), 8; right panel: WT, 8; β2(Ile275Ile), 4; β2(Ile275Iah), 6]. †For WT receptors, the reported unliganded open probabilities and free energies are upper and lower bounds, respectively, due to the limited ability to resolve small currents on the order of the noise.

desensitization, the observation of spontaneous unliganded opening in a WT background shows that elimination of the β2(Ile275:N-Pro273:O) H-bond strongly promotes channel opening. Thus, this H-bond stabilizes a non-conducting conformation to the extent that it is required to keep the unliganded channel closed with high probability.

### H-bond energetics

In the GoF background, it is straightforward to estimate the changes in unliganded $P_o$ upon ablation of the H-bond β2(Ile275:N-Pro273:O). Using Eq. 1, we calculate that these two bonds (one per β2 subunit) inhibit GoF channel opening by -1.8 kcal/mol (Fig. 3c middle). For WT or β2(Ile275Ile), we typically observed very small PTX-sensitive currents from which we estimate a basal $P_o$ of -0.002 (Fig. 3b right), similar to that reported by a previous study using the same method[39]. Based on this measure, we estimate that the two H-bonds β2(Ile275:N-Pro273:O) inhibit WT channel opening by -1.5 kcal/mol (Fig. 3c right), similar to our estimation for GoF channels. For WT receptors, the maximal $P_o$ upon activation by saturating GABA is -0.8[40], which given our estimate for basal $P_o$ implies that the chemical energy from GABA binding results in a shift of the closed-open free energy difference by -4.5 kcal/mol. Thus, breaking the main-chain H-bond β2(Ile275:N-Pro273:O) in each of two subunits accounts for -1/3ʳᵈ of the total activation energy in a WT receptor.

Importantly, the PTX-sensitive currents for WT were sufficiently small to be difficult to disambiguate from noise or solution exchange artifacts. Thus, our estimate of basal $P_o$ in WT channels represents an upper limit, and conversely, the calculated free energy changes based on this value reflect lower limits. Indeed, if we take another reported measure for WT basal $P_o$ of $1 \times 10^{-5}$[41], then our estimate for the energetic effect of ablation of this H-bond would be −5.3 kcal/mol, which is 70% of the −7.6 kcal/mol thereby estimated for the energy associated with maximal WT channel activation by GABA (i.e., $P_o$ = 0.8). We note that this measure of the total activation energy by GABA is similar to the −8.9 kcal/mol estimated from observations of the temperature dependence of GABA binding[42]. In summary, the two H-bonds β2(Ile275:N-Pro273:O) inhibit WT channel opening by at least the conservative estimate of -1.5 kcal/mol.

### H-bond in β2 is state-dependent

In the WT background, elimination of the H-bond β2(Ile275:N-Pro273:O) with the substitution β2(Ile275Iah) enhanced apparent GABA sensitivity by left-shifting GABA CRCs -50-fold (Fig. 3d right). This shows that the H-bond β2(Ile275:N-Pro273:O) is allosterically linked to the GABA binding site(s) in the ECD such that its ablation confers global conformational changes reminiscent of those occurring during activation by GABA—i.e., opening of the channel pore in the TMD and a conformational change of the ECD resulting in a higher average affinity for GABA. Thus, we hypothesize that normal channel activation by GABA involves breaking of the main-chain H-bond β2(Ile275:N-Pro273:O).

To further explore this idea, we examined the spatial proximity of the donor and acceptor atoms as a proxy for H-bond likelihood during all-atom molecular dynamics simulations of α1β2γ2 receptors. Comparison of simulations for antagonist-bound (i.e., closed) and GABA-bound (i.e., activated/desensitized) conformations indicates that channel activation is associated with an increased donor-acceptor separation for β2(Ile275:N-Pro273:O), consistent with a reduced H-bond likelihood in open versus closed states (Fig. 4a, b). In contrast, donor-acceptor distance distributions for the analogous H-bonds in α1 and γ2 subunits were relatively independent of ligation state and largely similar to that for β2 subunits in a closed state. If anything, H-bond distances in α1 and γ2 subunits were slightly shortened on average upon receptor activation by GABA. These results are consistent with our observation that the H-bond α1(Val279:N-Pro277:O) is not involved in channel function.

A recent simulated open state conformation[43] also shows longer distances on average for the discussed H-bond donor/acceptor pairs in β2 as compared to their analogous counterparts in α1 and γ2 subunits (Supplementary Fig. 6). Also, simulations of complexes with both GABA and potentiators including diazepam, etomidate, or propofol similarly recapitulate the observation that this H-bond is weaker (i.e., adopts a longer distance separation) in activated/desensitized complexes primarily in β2 subunits, whereas the analogous H-bond in the α1 subunit is independent of ligation state (Supplementary Fig. 7). Taken together with its allosteric linkage to the GABA binding site(s) in WT receptors, these simulations support the idea that specifically the main-chain H-bond β2(Ile275:N-Pro273:O) stabilizes a closed conformation and that this bond is broken or weakened during channel opening (Fig. 4c).

## Discussion

Here we show that a main-chain H-bond within the β2 M2-M3 linker inhibits pore opening, is required to keep the unliganded channel closed at rest, and that breaking of this bond, as we hypothesize occurs during activation by GABA, accounts for at least -1/3ʳᵈ of the energy derived from GABA binding. Structures suggest that this H-bond stabilizes a crimp or shallow hairpin in the M2-M3 linker backbone predicted to limit linker flexibility (Fig. 1b). We propose that increased flexibility of the β2 M2-M3 linker upon breaking this bond contributes to the physical basis for channel activation. Consistent with this idea, the substitution β2(Ile275Ala), which we predict to also increase linker flexibility by removing side-chain volume and reducing hydrophobicity near the center of the linker, has a similar functional signature: enhanced opening with a constitutive unliganded PTX-sensitive current and increased apparent GABA sensitivity[37].

Ablation of the analogous main-chain H-bond within the α1 M2-M3 linker had no effect on channel function. This also qualitatively parallels prior observations of asymmetric effects for alanine substitutions predicted to increase M2-M3 linker flexibility in α1, β2 or γ2 subunits[37]. Namely, β2(Ile275Iah) or β2(Ile275Ala) promote channel opening and prevent the channel from remaining closed, whereas the analogous

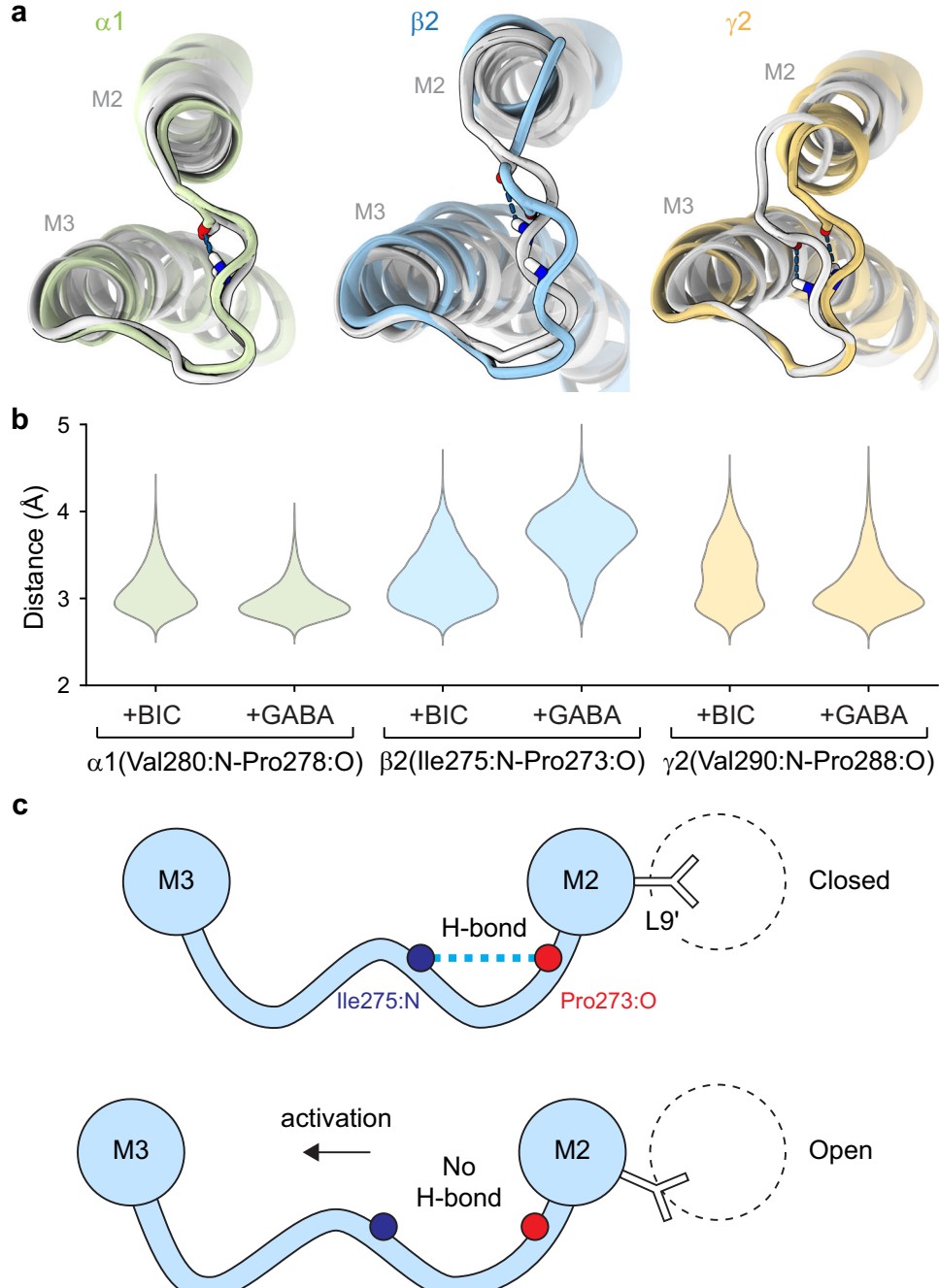

**Fig. 4 | A state-dependent main-chain H-bond in the β2 subunit M2-M3 linker is a hair-trigger for channel opening. a** Representative poses for M2-M3 linkers from MD simulations of antagonist-bound (i.e., closed; PDB 6X3S; white) and GABA-bound (i.e., activated; PDB 6X3Z; color) human GABA$_A$R complexes, aligned per subunit on local M2 and M3 helices. H-bonds α1(Val280:N-Pro278:O), β2(Ile275:N-Pro273:O) and γ2(Val290:N-Pro288:O) are indicated with dashed lines, and participating atoms are shown as sticks. **b** Distance distributions between donor and acceptor atoms for the H-bonds depicted in panel (**a**) from MD simulations of either antagonist-bound (+BIC, bicuculine) or GABA-bound (+GABA) complexes. Distributions are aggregate across four replicates of 500 ns simulations (Supplementary Fig. 5). Note, human α1(Val280:N-Pro278:O) is analogous with rat α1(Val279:N-Pro277:O) (Supplementary Fig. 1). **c** Cartoon illustrating the proposed breaking of the state-dependent H-bond β2(Ile275:N-Pro273:O) within the M2-M3 linker during channel opening.

mutations α1(Val279Vah), α1(Val279Ala), or γ2(Val290Ala) either have no effect or inhibit channel gating. Although other studies have observed subunit-specific effects of M2-M3 linker mutations[31,33], a coherent picture for the physical basis underlying this asymmetry is unclear. We hypothesize that linker flexibility is differentially transduced to channel gating at distinct subunits or interfaces. Asymmetry in the flexibility of M2 helices was also observed for cysteine cross-linking, which indicates that the pore-lining M2 helices in β subunits are more radially mobile than those in α subunits[44,45]. Consistent with these observations, comparison of antagonist- and GABA-bound cryo-EM structures show that activation involves relatively larger radial motions of M2-M3 linkers in β2 subunits as compared to α1 or γ2 subunits[23,34,35]. Taken together, these results strongly suggest that flexibility of the β2 subunit M2-M3 linkers has a special role in channel activation as compared to the linkers in α1 or γ2 subunits, possibly due to their physical location at the GABA-binding inter-subunit interfaces.

Static structures of synaptic GABA$_A$Rs enable a rough prediction of the main-chain H-bonds investigated here. However, not only do they not make clear their mechanistic importance, they also do not completely clarify their existence. Based on default distance and rotamer cutoffs in programs such as ChimeraX[46], the investigated H-bonds are identified in some distinct subunits within some structures, but not all subunits/structures. It is unclear to what extent this reflects a lack of resolution in more mobile loop regions and/or dynamics of these bonds. MD simulations indicate a preference for donor-acceptor distances of just below 3Å (except for β2 subunits with GABA bound) (Fig. 4a), typical for N·O H-bonds in proteins[47,48]. However, non-static donor-acceptor distance distributions exhibit tails extending to over 4Å, consistent with a dynamic nature to the electrostatic strength of these main-chain H-bonds, which may obfuscate their identification in static structures.

In conclusion, a main-chain H-bond within the β2 subunit M2-M3 linker is a critical component for stabilizing a resting-closed state of the channel. We hypothesize that this bond inhibits pore opening by limiting M2-M3 linker flexibility, and that breaking of this bond, or its conversion from stronger to weaker average strength, accounts for at least ~1/3$^{rd}$ of the activation energy supplied upon binding of GABA at both agonist sites during channel gating. Other main-chain interactions in gating loops may also be important aspects of the physical basis for the behavior of GABA$_A$Rs, as well as other pLGICs. However, their existence and/or contribution to channel function is typically not obvious from static structures and remains to be determined. Lastly, it is notable that this hair-trigger mechanism for rapid channel opening involves only main-chain atoms and, thus, is largely immune to genetic variation.

## Methods

### Ethical statement

*Xenopus laevis* frogs' care and surgery followed the ARRIVE guidelines and the University of Texas at Austin IACUC-approved protocol.

### Mutagenesis and in vitro transcription

DNA for wild-type and mutant GABA$_A$R rat α1, β2, and γ2 subunits was subcloned in the pUNIV vector[49]. The mature protein numeration for β2 and γ2 subunits is the same for rat and human, but the numeration for the rat α1 subunit is (human numeration −1) for most of the subunit (Supplementary Fig. 1). For ncAA incorporation, the codon of the residue of interest was replaced by the TAG stop codon (in contrast to TGA stop codons for each subunit). Mutations were introduced using QuikChange II (Qiagen) or by GenScript and confirmed by sequencing of the entire subunit. Sequences for all constructs are provided in supplementary information (Supplementary Data 1–6). Complementary RNA (cRNA) for each construct was generated (mMessage mMachine T7, Ambion), quantified (Qubit, ThermoFisher Scientific) and quality assessed (TapeStation, Agilent) prior to injection in *Xenopus laevis* oocytes.

### ncAA synthesis

For nonsense suppression in GABA$_A$ subunits, we used TAG mutants of the α1 and β2 GABA$_A$ subunits and PylT tRNAs in *Xenopus laevis* oocytes. PylT lacking the two terminal CA nucleotides was synthetized by Integrated DNA Technologies, Inc., folded and misacylated as previously described[6]. Leu-, Val-, α-hydroxy Leu- (Lah), and α-hydroxy Val- (Vah) pdCpA-substrates were synthesized according to published procedures[6].

### TEVC recording in oocytes

Mature female *Xenopus laevis* frogs were obtained from Nasco and housed in the University of Texas at Austin animal facility. *Xenopus laevis* oocytes were harvested from frogs under tricaine anesthesia. A piece of ovary was removed from the frog and placed in isolation media (108 mM NaCl, 2 mM KCl, 1 mM EDTA, 10 mM HEPES, pH = 7.5). Oocytes were manually isolated from the thecal and epithelial layers using forceps and then incubated in a collagenase buffer (0.5 mg/mL collagenase from *Clostridium histolytic*, 83 mM NaCl, 2 mM KCl, 1 mM MgCl$_2$, 5 mM HEPES) to remove the follicular layer. Oocytes were injected with 12 ng of total cRNA for α1, β2, and γ2 subunits (wild-type or mutants) in a 1:1:10 ratio[50] (Nanoject, Drummond Scientific). When injecting the TAG mutant of a subunit, 125 ng of tRNA was mixed with the cRNA encoding the GABA$_A$ subunits. Oocytes were incubated in a sterile incubation solution (88 mM NaCl, 1 mM KCl, 2.4 mM NaHCO$_3$, 19 mM HEPES, 0.82 mM MgSO$_4$, 0.33 mM Ca(NO$_3$)$_2$, 0.91 mM CaCl$_2$, 10,000 units/L penicillin, 50 mg/L gentamicin, 90 mg/L theophylline, and 220 mg/L sodium pyruvate, pH = 7.5) at 16 °C. Currents from expressed channels 1–3 days post-injection were recorded in two-electrode voltage clamp (Oocyte Clamp OC-725C, Warner Instruments), digitized using a PowerLab 4/30 system (ADInstruments) and recorded using LabChart 8 software (ADInstruments). Data was obtained from at least two different batches of oocytes for each experimental group. On one occassion, we observed appreciable responses in *Blank-injected oocytes, thus we discarded all recordings from that batch of oocytes.

Oocytes were held at −70 mV and perfused continuously (2 mL/min) with ND96 buffer (96 mM NaCl, 2 mM KCl, 1 mM CaCl$_2$, 1 mM MgCl$_2$, 5 mM HEPES, pH 7.5) or ND96 buffer containing picrotoxin (PTX), GABA, or diazepam (DZ). PTX was diluted from a 0.5 M stock solution in DMSO. DZ was diluted from a 10 mM stock solution in DMSO. GABA was diluted from 300 mM stock solution in water. The recording protocol was as follows: a 10 s pulse of PTX was followed by a series of 20–40 s pulses of increasing concentrations of either GABA or DZ and a final 10 s pulse of PTX. Pulses were sufficiently long to resolve the peak response, and inter-pulse intervals were 5–15 min to allow washout with buffer and currents to return to baseline. A representative trace is shown in Fig. 2c. Current traces were individually detrended in MATLAB 2024b (Mathworks) by substracting a spline fit to manually selected baseline regions in each trace. GABA or DZ concentration-response curves (CRCs) were fit with the Hill equation:

$$\frac{I}{I_{max}} = \frac{1}{1 + \left(\frac{EC_{50}}{[ligand]}\right)^{n_H}} \qquad (2)$$

where $I$ is the magnitude of the ligand-elicited current, $[ligand]$ is GABA or DZ concentration, $EC_{50}$ is the concentration eliciting a half-maximal response, and $n_H$ is the Hill slope.

For α1(Leu9'Thr)-containing receptors, we often observed that the spontaneous unliganded current decreased in magnitude with time in a nonlinear fashion, decreasing more rapidly at the beginning of the recording and reaching a nearly stable baseline in the latter period of the recording. This is unlikely to be accounted for by changes in leak current alone, as PTX consistently reduced the current to a similar level at the beginning and end of the recording. Thus, we assume that this rundown of the spontaneous current reflects a reduction in the pool of active channels at the membrane. Therefore, for α1(Leu9'Thr)-containing receptors, we first fit a spline to the spontaneous current baseline and subtracted this fit from the raw current, as mentioned above. To normalize out the time-dependent loss of active channels, we divided the baselined currents by the magnitude of the spline approximation of the spontaneous current baseline (i.e., our proxy for the number of active channels). Finally, we normalized the resulting detrended currents between the final PTX- and maximal GABA-elicited current levels as an approximation for open probability. See Supplementary Fig. 2 for a depiction of this protocol, which reasonably accounts for most of the observed rundown as evidenced by the very similar magnitude of PTX-elicited responses at the beginning and end of the detrended recording

despite not enforcing this a priori. Most importantly, this detrending has almost no effect on the relative magnitude of the final responses to saturating GABA and PTX at the end of the recording, where the baseline is largely stable, and thus does not affect our measure of unliganded open probability. The only effect this procedure has is to reduce the relative magnitude of current responses to low GABA concentrations towards the beginning of the recording, effectively reducing the foot of the concentration-response curves in a way that we believe better reflects the actual GABA sensitivity of these channels. Regardless, this adjustment is irrelevant for all our major conclusions.

### Statistical analysis
Summary data was analyzed using Prism 10 (GraphPad). Symbols and error bars are mean ± SEM, and box plots show median and interquartile intervals. Where applicable, we applied One-way Brown-Forsythe ANOVA (as the standard deviations were not equal among groups) followed by Dunnett's T3 multiple comparisons test. Irrespective of $P$-values, we focus only on relatively large effects.

### Molecular dynamics simulations
All-atom simulations in explicit solvent were deemed most appropriate to assess steady-state dynamics, given the relatively high precision and accuracy of atomistic interactions that can be captured compared to e.g., coarse-grained methods. MD simulations were as described previously[27]. Briefly, each structure of the α1β2γ2 GABA_A receptor in the presence of relevant ligands was placed in an MD simulation box with dimensions $127 \times 127 \times 163$ Å$^3$, embedded in a bilayer of 400 1-palmitoyl-2-oleoyl-sn-glycero-3-phosphocholine (POPC) molecules, and subsequently solvated with TIP3P water[51] and 150 mM NaCl in CHARMM-GUI[52] (Supplementary Table 3). The CHARMM36m forcefield[53] was used to describe the protein. Ligand parameters were generated with CGenFF in CHARMM-GUI, with additional optimization using quantum mechanics for ligands with high penalty scores[54]. Simulations were performed using GROMACS 2019.3[55]. The temperature was kept at 300 K using a velocity-rescaling thermostat[56]. Parrinello-Rahman pressure coupling[57] ensured constant pressure, the particle mesh Ewald algorithm[58] was used for long-range electrostatic interactions, and hydrogen-bond lengths were constrained using the LINCS algorithm[59]. After each system was energy-minimized, sequential 10 ns equilibration steps were performed with gradual release of position restraints on heavy, backbone, and Cα atoms. The bicuculline ligands were restrained during equilibration. Four replicates of ~500 ns unrestrained simulations were then generated, and frames were analyzed every 2 ns, for a total of 1000 samples (4 replicates × 250 frames) for each system. Local RMSDs for M2-M3 linkers indicate convergence of simulations in the region of interest, and among replicates with different initial velocities, within the timescale of atomistic simulations (Supplementary Fig. 4). Distances between hydrogen-bonding atoms were calculated using MDAnalysis[60].

Simulations in the presence of bicuculline, GABA, GABA + diazepam, GABA + etomidate, or GABA + propofol were initiated from corresponding cryo-EM structures (PDB IDs: 6X3S, 6X3Z, 6X3X, 6X3V, or 6X3T, respectively). Alternative analyses of these simulations were also reported previously[27]. All simulations are available on Zenodo[61].

Generation, validation, and simulations of a predicted model of the GABA-bound open state were reported previously[43], and are also available on Zenodo[62]. From five replicate 200 ns unrestrained simulations, frames were analyzed every ns, for a total of 1000 samples (5 replicates × 200 frames).

### Reporting summary
Further information on research design is available in the Nature Portfolio Reporting Summary linked to this article.

## Data availability
All data are available in the main text or the Supplementary Information. The source data underlying Figs. 2d, and 3b–d, and Supplementary Fig. 3b–d can be found in the Source Data file. Accession codes used in this study: 6X3S; 6X3Z; 6X3X; 6X3V; 6X3T. MD simulations files have been deposited in Zenodo [10.5281/zenodo.8142630]. Source data are provided with this paper.

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

## Acknowledgments

This research was supported by NIH grants R01GM148591 to M.P.G-O. and R35GM148239 to C.A.A., and by Swedish Research Council (VR) grants 2019-02433 and 2021-05806 to R.J.H. and E.L. Additional support was provided by the Sven & Lily Lawski Foundation to S.E.L., and by the Knut & Alice Wallenberg Foundation and Swedish e-Science Research Center to R.J.H. and E.L.

## Author contributions

M.P.G.-O. conceived and supervised this work. J.D.G. carried out, and C.A.A. supervised, the ncAA synthesis. C.M.B., M.P.G.-O., and C.A.A. established ncAA protocols for TEVC recordings. C.M.B. and N.G.D. carried out, and C.M.B. analyzed, the TEVC recordings. S.E.L. and Y.Z. carried out and analyzed, and R.J.H. and E.L. supervised the molecular dynamics simulations. C.M.B., M.P.G.-O., and S.E.L. produced the visualizations. C.M.B. and M.P.G.-O. wrote the manuscript.

## Competing interests

The authors declare no competing interests.
