## [Transparent Peer Review file · Nature Communications]

A single main-chain hydrogen bond required to keep GABA_A receptors closed

Corresponding Author: Dr Cecilia Borghese

Version 0:

Reviewer comments:

Reviewer #1

(Remarks to the Author)

In this manuscript, the authors identified mutations in the M2-M3 linkers that suggest linker flexibility that have asymmetric subunit-specific correlations with channel opening. They use non-canonical amino acids (ncAAs) to investigate the role of main-chain H-hydrogen bonds (H-bonds) that may stabilize the M2-M3 linkers. ncAAs with altered main-chain chemistry were used. In particular, alpha-hydroxy acids allowed amide-to-ester substitutions in the main-chain that ablate H-bonds with the amide nitrogen while leaving the side-chain unchanged. They then showed that a single main-chain H-bond within the b2 subunit M2-M3 linker inhibits pore opening and is required to keep the unliganded channel closed. They observed this in both a gain-of-function construct as well as the wild type pore configuration. Furthermore, breaking this H-bond could account for approximately one third of the energy used to open the channel during activation by GABA. In contrast, the analogous H-bond in the a1 subunit has no effect on gating. Their observations support the idea that channel opening may involve the state-dependent alteration of this specific main-chain H-bond within the b2 subunit M2-M3 linker.

This is an important topic that is addressed in an excellent manner, and is definitely worthy of publication once a few issues and questions can be addressed.

1. Figure 1 shows analyses of the M2-3 linker for 6X3X, the the GABA receptor with bound GABA and diazepam. Some of the authors in this group have published other constructs containing agents like propofol and etomidate bound in the transmembrane domain. Do these constructs demonstrate the same differential H-bond distances in their bound state? The authors should comment on this.

2. While it is clear that breaking the Hbond in question increases channel opening, that energy estimates of channel opening are in the range of those associated with this Hbond breakage, and that a lengthening of this bond is differentially seen in molecular dynamics simulations between open and closed states of the channel, how do we actually know that channel opening requires this bond to break? One could just as easily say that a door opens more easily with a broken hinge, but that does not necessary mean that the hinge must be broken for the door to open.

The Lindahl Group, also authors on this paper, recently published a manuscript in ScienceAdvances in which they used a goal-oriented adaptive sampling strategy in molecular dynamics simulations followed by Markov state modeling to capture an energetically stable putative open state of the receptor. Can they comment on the status of this Hbond that they see in their model of the open channel?

What about the same type of intrachain Hbonds between TYR 277:O and LYS 279:N, or any other intrachain Hbond possibilities in that loop?

page 5 line 94: the authors use the prime nomenclature here while starting out with the initial amino acid residue numbers elsewhere. It would be helpful to have the actual amino acid number for that Leu9' for reference to actual models.

page 6 line 131: I think that "First, b2(Ile275Ile) recapitulated...: should have b2(Ile275Ile*)?"

(Remarks to the Author)

The manuscript by the Goldschen-Ohm group (in collaboration with those of Chris Ahern's and Erik Lindahl's) describes the effect of selectively preventing the formation of a main-chain hydrogen bond (H-bond) in the M2-M3 linker of the $\alpha 1$ or $\beta 2$ subunits of a heterotrimeric GABAAR on GABA-evoked currents. The manuscript provides enough detail in the methods for the work to be reproduced. A most obvious strength of the paper is the use of unnatural amino-acid mutagenesis to replace an amide peptide bond with an ester without affecting the side chain. Although already employed in numerous other cases (including members of the Cys-loop receptor superfamily), this type of non-classical mutagenesis remains under-utilized. The main conclusion of the manuscript is that preventing the formation of a main-chain hydrogen bond in the M2-M3 linker has a gain-of-function effect that manifests as an increased open probability (P_o) in the absence of agonist ("unliganded P_o "). The effect of this mutation (in two subunits) is too small, however, to be properly quantified when engineered in the wild-type background, and hence, the quantification was performed in the background of another gain-of-function mutation in the middle of the M2 α -helix that also increases the unliganded P_o . The effect of the latter mutations is so high that, under these conditions, the effect of the amide-to-ester mutation can be appreciated more clearly and quantified with more certainty. The authors argue that structures cannot reveal the presence or absence of hydrogen bonds with confidence, and hence, perform, all-atom MD simulations of cryo-EM models of the antagonist-bound (closed) and GABA-bound (desensitized) conformations to infer the state of the hydrogen bonds in question (formed or broken, strong or weak) in the closed versus open states. The results are interpreted as an indication that the main-chain H-bond involving $\beta 2$ Ile-275 breaks during the closed \rightarrow open conformational change, whereas those involving the aligned position of the $\alpha 1$ subunits (Val-279) do not.

As could be expected from the assembled team of experts (a genuine "dream team" of ion-channel biophysicists), I find the execution of the experiments to be flawless. Having said this, however, I also find a few aspects of the paper that could be modified to make the presentation clearer and to limit the extent of the conclusions. Below, I present my observations in the order in which they appear:

Line 54: I think reference 19 is incorrect. In Sente et al (2022), the solved structures correspond to extra-synaptic $\alpha 4$ -containing heterotrimeric and β - γ/δ heterodimeric GABAARs (not $\alpha 1$ -containing receptors, as stated in the text, though).

Lines 61-64: It is not clear to me how the "comparison of structural snapshots" in unliganded and GABA-bound conformations can suggest that the rearrangement of the M2-M3 linkers "sequentially follow(s) the M2 pore-lining helices". Which one follows which? And how can one infer a temporal sequence of rearrangements between two conformations from the observation of only two structures?

Line 122: The effect of the Ile275Iah mutation in the $\beta 2$ subunits on the unliganded P_o is described as being "massive". It should probably be stressed that this is only because the unnatural amino-acid mutation was engineered in the background of another gain-of-function mutation that brings the unliganded P_o to a highly sensitive region of the " P_o space". In the wild-type background, on the other hand, the unliganded P_o of the $\beta 2$ Ile275Iah mutant is actually rather small (~ 0.07 ; Fig. 3b, right). And if a different background mutation that increases the unliganded P_o to, say, 0.95, had been used instead of Leu9⁺Thr, the observed effect of the $\beta 2$ Ile275Iah mutation would also have seemed to be very small (it could not have exceeded 0.05). In other words, the observed "massive" effect is a consequence of how ion-channel function is being probed. Taking the value of ~ 1.8 kcal/mol (line 123), one can calculate that the Ile275Iah mutation in both $\beta 2$ subunits increases the unliganded equilibrium constant by a factor of ~ 20 , that is, a factor of $\sim 4.5^2 \approx 20$. In my opinion, "a factor of 4.5" is a far more objective way to describe the effect of the Ile275Iah mutation. To put things in context, the leucine-to-threonine mutation at position 9⁺ of M2 used in the background increases the unliganded P_o by a factor that is even larger than 4.5 (Fig. 3b, middle panel, GoF compared to Fig. 3b, right panel, Iah).

Line 123: The authors state "... this H-bond naturally inhibits channel opening by ~ 1.8 kcal/mol". This calculation uses the values of unliganded P_o of the background M2 mutant and the combined ($\beta 2$ Ile275Iah + M2) mutant. The implicit assumption, here, is that the effect of these two gain-of-function mutations are additive, which would be perfectly fine. However, a few lines below, (line 126), the authors entertain the possibility that the effects of these two mutations are NOT additive. This is a bit confusing. If they were not additive, then the 1.8 kcal/mol value would not mean much. This numerical value would depend on the background construct, and the only background construct that makes sense is the wild-type, for which this calculation could not be performed with confidence.

Line 124: It is not clear why the increase in unliganded P_o caused by the $\beta 2$ Ile275Iah mutation does not result in a leftward displacement of the concentration-response curve when engineered in the background of the M2 mutation (Fig. 3d, middle). To address this unexpected finding, the authors argue that the Ile275Iah mutation and the M2 mutation "can have non-additive effects". Yes, I understand that the combination need not be additive, but if the Ile275Iah mutation increases the unliganded P_o of the background M2 mutant, then I (perhaps, naively) also expect a shift of the concentration-response curve (Figure 3d, middle) to the left, whether the effects of the mutations are perfectly additive or not.

Figure 3:

It is unclear how the upper-limit value of the unliganded P_o of the wild-type channel was estimated. In the second-from-the-right panel of part **a**, PTX does not seem to reduce the currents at all. Hence, with an observed zero value for the unliganded P_o , the unliganded gating equilibrium constant could be arbitrarily low. It is not clear, then, how the value of 0.002 (line 140) was estimated. In part **b**, leftmost and middle panels: What is the difference between the respective "GoF" values? I thought they were the same thing, but evidently, I am wrong. In part **d**, I think the three panels would be most clearly displayed using

the same range for the concentration axes.

Line 142: The value of ~1.5 kcal/mol calculated for the $\beta 2$ Ile275Iah mutant engineered in the wild-type background is said to be similar to the value of ~1.8 kcal/mol calculated for the same mutant engineered in the background of the M2 mutation. Once again: Are the effects additive, then? And do these numbers matter after all? Are they necessary?

Lines 150-153: Please, write this section more clearly. Perhaps, less succinctly.

Line 157: The MD simulations compared a model of an antagonist-bound (closed) conformation to a model of a GABA-bound desensitized conformation. However, the electrophysiological recordings report on the occupancy of the open state. I understand that an atomic model of an open $\alpha 1\beta 2\gamma 2$ GABAAR may not be available in the literature yet, but can we use a model of a desensitized state and “pretend” it is an open state? Wouldn't the recently simulated model of a GABAAR in the open state (Haloi et al, 2025) be a more attractive choice for this simulation, especially considering that it originated in the Lindahl lab?

Line 177: In the Results section, the possibility that a main-chain H-bond involving $\beta 2$ Ile-275 breaks upon opening and forms again upon closing is “timidly” proposed as a possibility hinted by the data. Considering the simulation results, I find the timid tone very appropriate. In fact, the authors admit that the evidence for this phenomenon (inferred from the simulations) is rather tenuous. The Discussion, however, boldly assumes that the breaking/formation of this bond is part of the closed-open conformational change, and the same seems to be the case for the last few sentences of the Abstract. I fail to follow the reasoning that connects the observed effect of the amide-to-ester mutation on the unliganded closed-open equilibrium to the breaking of the H-bond in question upon opening of the wild-type channel. The H-bond may well be formed in both, the closed and open states of the wild-type GABAAR, and its disruption caused by the Ile-to-Iah mutation may simply affect the closed and open conformations differently, stabilizing the latter relative to the former. The structural consequences of the Ile-to-Iah mutation may extend beyond the highly local, mere elimination of a main-chain H-bond. Indeed, proteins are expected to “relax” to new conformations to accommodate mutations. Thus, in the absence of structures of the Ile-to-Iah mutant in the closed and open conformations, perhaps, one should remain very cautious.

Version 1:

Reviewer comments:

Reviewer #1

(Remarks to the Author)

The authors have thoroughly and satisfactorily addressed my comments and suggestions.

Reviewer #2

(Remarks to the Author)

The authors have satisfactorily addressed most of my concerns. However, I still feel that the evidence for the M2-M3 loop hydrogen bond forming and breaking upon gating ONLY comes from the MD simulations. The functional assays ONLY show that the H-bond in question is required to keep the unliganded gating equilibrium constant (and by extension, the gating equilibrium constants of the agonist-bound channel, as well) at physiologically meaningful values. But these results do not suggest (let alone, indicate) that the bond **forms and breaks** as the GABA-bound receptor changes conformation. Let us imagine, for example, a disulfide bond. I can imagine that mutating either cysteine to, say, a serine, will (de)stabilize the closed, open and desensitized states to different degrees. Hence, the gating equilibrium constants of the mutant will be affected, potentially, a lot. Does this result mean that, in the wild-type construct, the disulfide forms and breaks as the channel goes back-and-forth between these different conformations? I am afraid not. In my (very humble) opinion, this is precisely the case for the M2-M3 loop H-bond of this manuscript (the covalent versus non-covalent difference between a disulfide bond and a hydrogen bond is irrelevant for this reasoning). The evidence for the H-bond forming/breaking (or becoming stronger/weaker) comes from the observation of structures. In the absence of structures of the mutant (which I am not expecting!), MD seems to be the next best thing. In the last sentence of the Abstract, “Our *observations* support ...” could perhaps be changed to “Our *molecular simulations* support ...”.

REVIEWER COMMENTS

We thank the reviewers for their careful evaluation and constructive comments which we believe have helped to improve this work.

FIGURE CHANGES

- **Figure 3d:** Adjusted x-axes so that each panel has the same range.
- **Extended Data Figure 1:** Highlighted location of the main pore gate 9' leucine.
- **Extended Data Figure 4:** Added RMSD for simulated open state.
- **NEW Extended Data Figure 6:** H-bond distance distributions for a simulated open state.
- **NEW Extended Data Figure 7:** H-bond distance distributions for various ligation complexes.

Reviewer #1 (Remarks to the Author):

In this manuscript, the authors identified mutations in the M2-M3 linkers that suggest linker flexibility that have asymmetric subunit-specific correlations with channel opening. They use non-canonical amino acids (ncAAs) to investigate the role of main-chain H-hydrogen bonds (H-bonds) that may stabilize the M2-M3 linkers. ncAAs with altered main-chain chemistry were used. In particular, alpha-hydroxy acids allowed amide-to-ester substitutions in the main-chain that ablate H-bonds with the amide nitrogen while leaving the side-chain unchanged. They then showed that a single main-chain H-bond within the b2 subunit M2-M3 linker inhibits pore opening and is required to keep the unliganded channel closed. They observed this in both a gain-of-function construct as well as the wild type pore configuration. Furthermore, breaking this H-bond could account for approximately one third of the energy used to open the channel during activation by GABA. In contrast, the analogous H-bond in the a1 subunit has no effect on gating. Their observations support the idea that channel opening may involve the state-dependent alteration of this specific main-chain H-bond within the b2 subunit M2-M3 linker.

This is an important topic that is addressed in an excellent manner, and is definitely worthy of publication once a few issues and questions can be addressed.

1. Figure 1 shows analyses of the M2-3 linker for 6X3X, the the GABA receptor with bound GABA and diazepam. Some of the authors in this group have published other constructs containing agents like propofol and etomidate bound in the transmembrane domain. Do these constructs demonstrate the same differential H-bond distances in their bound state? The authors should comment on this.

This is an interesting question. First, it is important to note that this H-bond is not reliably resolved in any of the structures. Using default H-bond criteria in ChimeraX, this H-bond is identified here and there in some of the structures, but even then, only here and there in some of the subunit chains. However, even when not identified using default criteria, the atoms are not too far off and thus we initially suspected that an H-bond might be present.

To better address this question, we have turned to MD simulations initiated from structures in various ligation states, where we can explore the dynamic distance distribution that the H-bond N and O atoms may sample. We now show such distance distributions from MD simulations for structures with GABA + diazepam, GABA + etomidate, or GABA + propofol bound in Extended Data Fig. 7. To summarize, these simulations recapitulate the observation that this H-bond adopts a longer average distance when agonist/agonist + modulator are bound versus bound antagonist in beta2 subunits, whereas the analogous H-bond in alpha1 subunits is invariant to ligation state. There is also some ligation state-dependence to this bond in simulations for the gamma2 subunit, but nowhere near as clear of a shift as seen in beta2 subunits.

2. While it is clear that breaking the Hbond in question increases channel opening, that energy estimates of channel opening are in the range of those associated with this Hbond breakage, and that a lengthening of this bond is differentially seen in molecular dynamics simulations between open and closed states of the channel, how do we actually know that channel opening requires this bond to break? One could just as easily say that a door opens more easily with a broken hinge, but that does not necessary mean that the hinge must be broken for the door to open.

Fundamentally we agree with the reviewer that perturbations that increase channel opening could have nothing to do with normal gating. However, we believe that breaking/weakening of this H-bond is likely to be part of normal gating for two reasons:

1) Breaking the H-bond in a WT background increased apparent affinity for GABA (Fig. 3d right), which shows that it induces global conformational changes affecting the GABA binding site(s) that are consistent with what would be expected for normal channel activation. We have attempted to better explain this reasoning based on our observed shift in the GABA CRC (lines 172-175). The idea that mutations eliciting spontaneous opening do so by the same mechanism as agonists has also been used successfully to account for mutations and drug modulation in GABA_A receptors (e.g., PMC7899671, PMC7052843, PMID 15791108).

2) MD simulations show a ligation-dependent shift in the H-bond distance consistent with breaking or weakening of this H-bond in activated vs non-activated ligation states. Also see new data in Extended Data Figs. 6-7 including a simulated open state. This is also consistent with larger observed relative changes in beta subunit M2-M3 linkers between cryo-EM structures in non-activated vs. activated ligation states as compared those in alpha or gamma subunits.

We believe that this evidence reasonably suggests that this H-bond is part of the normal gating mechanism. Nonetheless, we agree that we cannot completely rule out the possibility that despite “quacking” like normal channel gating, the observed opening upon H-bond ablation could be abnormal. As such, we have edited several of our statements in the discussion to emphasize that this is our prediction.

3. The Lindahl Group, also authors on this paper, recently published a manuscript in ScienceAdvances in which they used a goal-oriented adaptive sampling strategy in molecular dynamics simulations followed by Markov state modeling to capture an energetically stable putative open state of the receptor. Can they comment on the status of this Hbond that they see in their model of the open channel?

We now show H-bond distance distributions for all subunits in the simulated open state model in Extended Data Fig. 6. To summarize, simulations of this model offer further evidence that channel opening (as well as desensitizing) releases the H-bond of interest (shifts it to longer N-O distances) specifically in $\beta 2$. Interestingly, this shift was primarily evident in one of the two $\beta 2$ subunits in each replicate, possibly reflecting the more pronounced asymmetry of the open versus desensitized states, and/or of the computational model versus experimental structures. Nonetheless, the open model supports the central identification of a state-dependent change in a $\beta 2$ -subunit main-chain H-bond.

4. What about the same type of intrachain Hbonds between TYR 277:O and LYS 279:N, or any other intrachain Hbond possibilities in that loop?

We are very interested in other H-bonds in this region, and we are pursuing projects involving them. We would prefer to reserve these experiments for a future study.

5. page 5 line 94: the authors use the prime nomenclature here while starting out with the initial

amino acid residue numbers elsewhere. It would be helpful to have the actual amino acid number for that Leu9' for reference to actual models.

Done (lines 95-96).

6. page 6 line 131: I think that "First, b2(Ile275Ile) recapitulated...: should have b2(Ile275Ile*)?"

Throughout the manuscript, the asterisk represents either the TAG stop codon or the tRNA that recognizes this stop codon. Therefore Ile275* plus *Ile, *Iah or *Blank results in Ile275Ile, Ile275Iah or Ile275Blank, respectively. So beta2(Ile275Ile) represents a beta2 subunit where the Ile275 was replaced by a TAG stop codon, and coinjected with a UAG-directed-tRNA-Ile, resulting in an Ile in position 275, that is, a wild-type beta2 subunit. We are sorry for any confusion that may arise, but we arrived at this notation after many long discussions, and we believe it is the simplest one possible. We modified the captions (Figs. 2, 3; Extended Data Fig. 3) and simplified the methods (lines 445-446) to improve readability.

Reviewer #2 (Remarks to the Author):

The manuscript by the Goldschen-Ohm group (in collaboration with those of Chris Ahern's and Erik Lindahl's) describes the effect of selectively preventing the formation of a main-chain hydrogen bond (H-bond) in the M2-M3 linker of the $\alpha 1$ or $\beta 2$ subunits of a heterotrimeric GABAAR on GABA-evoked currents. The manuscript provides enough detail in the methods for the work to be reproduced. A most obvious strength of the paper is the use of unnatural amino-acid mutagenesis to replace an amide peptide bond with an ester without affecting the side chain. Although already employed in numerous other cases (including members of the Cys-loop receptor superfamily), this type of non-classical mutagenesis remains under-utilized. The main conclusion of the manuscript is that preventing the formation of a main-chain hydrogen bond in the M2-M3 linker has a gain-of-function effect that manifests as an increased open probability (P_o) in the absence of agonist ("unliganded P_o "). The effect of this mutation (in two subunits) is too small, however, to be properly quantified when engineered in the wild-type background, and hence, the quantification was performed in the background of another gain-of-function mutation in the middle of the M2 α -helix that also increases the unliganded P_o . The effect of the latter mutations is so high that, under these conditions, the effect of the amide-to-ester mutation can be appreciated more clearly and quantified with more certainty. The authors argue that structures cannot reveal the presence or absence of hydrogen bonds with confidence, and hence, perform, all-atom MD simulations of cryo-EM models of the antagonist-bound (closed) and GABA-bound (desensitized) conformations to infer the state of the hydrogen bonds in question (formed or broken, strong or weak) in the closed versus open states. The results are interpreted as an indication that the main-chain H-bond involving $\beta 2$ Ile-275 breaks during the closed \rightarrow open conformational change, whereas those involving the aligned position of the $\alpha 1$ subunits (Val-279) do not.

As could be expected from the assembled team of experts (a genuine "dream team" of ion-channel biophysicists), I find the execution of the experiments to be flawless. Having said this, however, I also find a few aspects of the paper that could be modified to make the presentation clearer and to limit the extent of the conclusions. Below, I present my observations in the order in which they appear:

We thank the reviewer for their praise and insightful comments.

1. Line 54: I think reference 19 is incorrect. In Sente et al (2022), the solved structures correspond to extra-synaptic $\alpha 4$ -containing heterotrimeric and β - γ/δ heterodimeric GABAARs (not $\alpha 1$ -containing receptors, as stated in the text, though).

The reviewer is correct that the main object of the referenced study was $\alpha 4$ -containing heterotrimeric and β - γ/δ heterodimeric GABA_ARs. However, the study also provided a structure of $\alpha 1\beta 3\gamma 2 + \text{Ro}15-4513$ (PDB: 7QNE), which is why we included it in the list of references. Nonetheless, we have removed this reference where we cite $\alpha\beta\gamma$ structures to avoid confusion (line 54). We have retained the reference for the preceding sentence referring to general subunit stoichiometry.

2. Lines 61-64: It is not clear to me how the “comparison of structural snapshots” in unliganded and GABA-bound conformations can suggest that the rearrangement of the M2-M3 linkers “sequentially follow(s) the M2 pore-lining helices”. Which one follows which? And how can one infer a temporal sequence of rearrangements between two conformations from the observation of only two structures?

By “sequentially follow”, we were referring to the polypeptide sequence and not a temporal sequence. We apologize for the confusion our wording conferred, and we have amended the text to clarify (new lines 63-64).

3. Line 122: The effect of the Ile275Iah mutation in the $\beta 2$ subunits on the unliganded P_o is described as being “massive”. It should probably be stressed that this is only because the unnatural amino-acid mutation was engineered in the background of another gain-of-function mutation that brings the unliganded P_o to a highly sensitive region of the “ P_o space”. In the wild-type background, on the other hand, the unliganded P_o of the $\beta 2$ Ile275Iah mutant is actually rather small (~ 0.07 ; Fig. 3b, right). And if a different background mutation that increases the unliganded P_o to, say, 0.95, had been used instead of Leu9'Thr, the observed effect of the $\beta 2$ Ile275Iah mutation would also have seemed to be very small (it could not have exceeded 0.05). In other words, the observed “massive” effect is a consequence of how ion-channel function is being probed. Taking the value of ~ 1.8 kcal/mol (line 123), one can calculate that the Ile275Iah mutation in both $\beta 2$ subunits increases the unliganded equilibrium constant by a factor of ~ 20 , that is, a factor of ~ 4.5 per mutation ($4.5^2 \cong 20$). In my opinion, “a factor of 4.5” is a far more objective way to describe the effect of the Ile275Iah mutation. To put things in context, the leucine-to-threonine mutation at position 9' of M2 used in the background increases the unliganded P_o by a factor that is even larger than 4.5 (Fig. 3b, middle panel, GoF compared to Fig. 3b, right panel, Iah).

We agree with the reviewer. By using the subjective term “massively” we let our excitement over the observed phenomenon show. We have now removed this subjective term in favor of a quantitative measure. Although it may be irrelevant, we briefly describe our thinking which led us to the original text: In a GoF background, we estimate that this H-bond contributes ~ 2 kcal/mol to the pore equilibrium (~ 1 kcal/mol per H-bond in each of two beta2 subunits). In a WT channel we measure a similar energy contribution. However, our estimate in WT is a lower limit, and could easily be several fold higher. Even if we take this lower limit, what impressed us is not that this is a crazy energy for a H-bond, but that this energy is efficiently transduced into channel current, and it accounts for at least $\sim 1/3$ of the activation energy of a WT receptor. In our opinion, this is impressive for a single H-bond outside of either the pore or the agonist binding site. Nonetheless, we fully agree that the term “massively” is subjective, and we have removed it in favor of quantitative measures (lines 123-130).

4. Line 123: The authors state “... this H-bond naturally inhibits channel opening by ~ 1.8 kcal/mol”. This calculation uses the values of unliganded P_o of the background M2 mutant and the combined ($\beta 2$ Ile275Iah + M2) mutant. The implicit assumption, here, is that the effect of these two gain-of-function mutations are additive, which would be perfectly fine. However, a few lines below, (line 126), the authors entertain the possibility that the effects of these two mutations are NOT additive. This is a bit confusing. If they were not additive, then the 1.8 kcal/mol value would not mean much. This numerical value would depend on the background construct, and the only background construct

that makes sense is the wild-type, for which this calculation could not be performed with confidence.

This is an important point that we feel we did not clarify sufficiently. To address this, we have reworked our discussion of the energetics in GoF and WT backgrounds in a new section “H-bond energetics” (lines 145-167). To briefly summarize a few key points:

First, we would like to clarify that although we are uncertain as to the resting spontaneous open probability in WT channels, our estimates reflect an upper limit (e.g., some of the very small responses to PTX we have interpreted as block of spontaneously open channels could be artifact from noise or solution exchange over the oocyte). As such, the free energy change we measure upon H-bond disruption with beta2(Ile275Iah) in a WT background is a lower limit. Thus, although we cannot be extremely confident in the energetic measure of ~1.5 kcal/mol, it certainly isn't smaller than this in a WT channel. As such, we would suggest that our measures in the WT background are very insightful and valuable, and clearly show that in a WT channel breaking this H-bond accounts for at least ~1/3 of the activation energy. Indeed, if we overestimated the basal P_o in WT, then we would have underestimated both the magnitude of the H-bond's energetic contribution to gating and the fractional amount of the total activation energy that it accounts for, potentially significantly (see new text).

Second, based on our lower limit conservative estimate in WT, we do indeed observe approximately the same energetic effect of H-bond ablation in either GoF or WT backgrounds. However, if this were the case (i.e., the GoF and beta2(Ile275Iah) perturbations were additive) then we would expect, as apparently did the reviewer, to see a shift in the GABA CRC in the GoF background. Given that we did not observe this shift, we suspect that these perturbations are not entirely additive, which implies that our conservative estimate is likely too small and the actual energetic contribution in WT channels is larger than that measured in GoF channels. However, see our response to the next comment #5 as well.

5. Line 124: It is not clear why the increase in unliganded P_o caused by the $\beta 2$ Ile275Iah mutation does not result in a leftward displacement of the concentration-response curve when engineered in the background of the M2 mutation (Fig. 3d, middle). To address this unexpected finding, the authors argue that the Ile275Iah mutation and the M2 mutation “can have non-additive effects”. Yes, I understand that the combination need not be additive, but if the Ile275Iah mutation increases the unliganded P_o of the background M2 mutant, then I (perhaps, naively) also expect a shift of the concentration-response curve (Figure 3d, middle) to the left, whether the effects of the mutations are perfectly additive or not.

We agree with the reviewer that this result is not easily explained solely by changes in the closed-open equilibrium. It is worth noting that the predicted shift in the GoF background in an MWC model that assumes the effect is solely on the closed-open equilibrium alone is much smaller than the shift in a WT background or the shift due to the GoF mutation alone. Nonetheless, we still would expect to see at least some small shift of the CRC. It's possible that this combination of perturbations has influenced channel gating in a less simple way, for example by altering the closed and/or open state microscopic affinities for GABA. Ultimately, we cannot say for certain what underlies this observation. We have edited the text (lines 126-130) to better convey this uncertainty.

6. Figure 3:

It is unclear how the upper-limit value of the unliganded P_o of the wild-type channel was estimated. In the second-from-the-right panel of part **a**, PTX does not seem to reduce the currents at all. Hence, with an observed zero value for the unliganded P_o , the unliganded gating equilibrium constant could be arbitrarily low. It is not clear, then, how the value of 0.002 (line 140) was estimated. In part **b**, leftmost and middle panels: What is the difference between the respective “GoF” values? I thought

they were the same thing, but evidently, I am wrong. In part **d**, I think the three panels would be most clearly displayed using the same range for the concentration axes.

There is typically a tiny reduction in current magnitude upon PTX application, which we use to estimate P_o in the same way as for mutants with more constitutive basal activity. In our opinion, it is unclear to what extent this reduction is true block of spontaneously active channels or artifact from solution exchange across the oocyte, and thus represents an upper limit estimate for unliganded P_o in WT channels. Our estimate is approximately the same as that obtained by PMID 12969744 using the same approach. We now clarify this in the text as well as discuss an alternative estimate (lines 158-167).

In Fig. 3b-c the GoF values on the left versus middle panels reflect separate GoF recordings from the same batches of oocytes used to obtain the respective mutant recordings. We have amended the figure caption to clarify this (lines 409-410).

We have adjusted Fig. 3d so that all plots share the same concentration range.

7. Line 142: The value of ~ 1.5 kcal/mol calculated for the $\beta 2$ Ile275Iah mutant engineered in the wild-type background is said to be similar to the value of ~ 1.8 kcal/mol calculated for the same mutant engineered in the background of the M2 mutation. Once again: Are the effects additive, then? And do these numbers matter after all? Are they necessary?

See our response to comment #4 above.

8. Lines 150-153: Please, write this section more clearly. Perhaps, less succinctly.

We have attempted to clarify this section (new lines 172-175).

9. Line 157: The MD simulations compared a model of an antagonist-bound (closed) conformation to a model of a GABA-bound desensitized conformation. However, the electrophysiological recordings report on the occupancy of the open state. I understand that an atomic model of an open $\alpha 1\beta 2\gamma 2$ GABAAR may not be available in the literature yet, but can we use a model of a desensitized state and “pretend” it is an open state? Wouldn't the recently simulated model of a GABAAR in the open state (Haloi et al, 2025) be a more attractive choice for this simulation, especially considering that it originated in the Lindahl lab?

We agree with the reviewer, and we refer them to our response to Reviewer #1's comment #3 which addresses this question.

10. Line 177: In the Results section, the possibility that a main-chain H-bond involving $\beta 2$ Ile-275 breaks upon opening and forms again upon closing is “timidly” proposed as a possibility hinted by the data. Considering the simulation results, I find the timid tone very appropriate. In fact, the authors admit that the evidence for this phenomenon (inferred from the simulations) is rather tenuous. The Discussion, however, boldly assumes that the breaking/formation of this bond is part of the closed-open conformational change, and the same seems to be the case for the last few sentences of the Abstract. I fail to follow the reasoning that connects the observed effect of the amide-to-ester mutation on the unliganded closed-open equilibrium to the breaking of the H-bond in question upon opening of the wild-type channel. The H-bond may well be formed in both, the closed and open states of the wild-type GABAAR, and its disruption caused by the Ile-to-Iah mutation may simply affect the closed and open conformations differently, stabilizing the latter relative to the former. The structural consequences of the Ile-to-Iah mutation may extend beyond the highly local, mere elimination of a main-chain H-bond. Indeed, proteins are expected to “relax” to new conformations to

accommodate mutations. Thus, in the absence of structures of the Ile-to-Iah mutant in the closed and open conformations, perhaps, one should remain very cautious.

Please see our response to Reviewer #1's comment #2 which addresses this question.

REVIEWERS' COMMENTS AND REPLIES

Reviewer #1 (Remarks to the Author):

The authors have thoroughly and satisfactorily addressed my comments and suggestions.

Reviewer #2 (Remarks to the Author):

The authors have satisfactorily addressed most of my concerns. However, I still feel that the evidence for the M2-M3 loop hydrogen bond forming and breaking upon gating **ONLY** comes from the MD simulations. The functional assays **ONLY** show that the H-bond in question is required to keep the unliganded gating equilibrium constant (and by extension, the gating equilibrium constants of the agonist-bound channel, as well) at physiologically meaningful values. But these results do not suggest (let alone, indicate) that the bond **forms and breaks** as the GABA-bound receptor changes conformation. Let us imagine, for example, a disulfide bond. I can imagine that mutating either cysteine to, say, a serine, will (de)stabilize the closed, open and desensitized states to different degrees. Hence, the gating equilibrium constants of the mutant will be affected, potentially, a lot. Does this result mean that, in the wild-type construct, the disulfide forms and breaks as the channel goes back-and-forth between these different conformations? I am afraid not. In my (very humble) opinion, this is precisely the case for the M2-M3 loop H-bond of this manuscript (the covalent versus non-covalent difference between a disulfide bond and a hydrogen bond is irrelevant for this reasoning). The evidence for the H-bond forming/breaking (or becoming stronger/weaker) comes from the observation of structures. In the absence of structures of the mutant (which I am not expecting!), MD seems to be the next best thing. In the last sentence of the Abstract, “Our *observations* support ...” could perhaps be changed to “Our *molecular simulations* support ...”.

Reply to reviewers

We thank the reviewers for their thorough and positive criticism of our work. We believe it greatly improved the manuscript.

We modified the last sentence of the abstract according to Reviewer #2's suggestion.

Other changes (highlighted) are to further clarify some points raised by the editorial team and due to the journal's formatting guidelines.